# Advantages of Amending Chemical Fertilizer with Plant-Growth-Promoting Rhizobacteria under Alternate Wetting Drying Rice Cultivation

Chesly Kit Kobua [1] , Ying-Tzy Jou [2,*] and Yu-Min Wang [3]

1    Department of Tropical Agriculture and International Cooperation, National Pingtung University of Science and Technology, Pingtung County 91201, Taiwan; chesly.kobua@gmail.com
2    Department of Biological Science and Technology, National Pingtung University of Science and Technology, Pingtung County 91201, Taiwan
3    General Research Service Center, National Pingtung University of Science and Technology, Pingtung County 91201, Taiwan; wangym@mail.npust.edu.tw
*    Correspondence: ytjou@mail.npust.edu.tw; Tel.: +886-8-770-3202 (ext. 6365); Fax: +886-8-7740550

**Abstract:** Chemical fertilizer (CF) is necessary for optimal growth and grain production in rice farming. However, the continuous application of synthetic substances has adverse effects on the natural environment. Amending synthetic fertilizer with plant-growth-promoting rhizobacteria (PGPR) is an alternate option to reduce CF usage. In this study, a field trial was undertaken in southern Taiwan. We aimed to investigate the effects of reducing CF, either partially or completely, with PGPR on the vegetative growth, biomass production, and grain yield of rice plants cultivated under alternate wetting and drying (AWD) cultivation. In addition, we aimed to determine an optimal reduction in CF dose when incorporated with PGPR for application in rice cultivation under AWD. The trial consisted of four treatments, namely, 0% CF + 100% PGPR (FP1), 25% CF + 75% PGPR (FP2) 50% CF + 50% PGPR (FP3), and 100% CF + 0% PGPR (CONT). A randomized complete blocked design (RCBD) with three replicates was used. A reduction in CF by 25–50% with the difference compensated by PGPR significantly ($p \leq 0.05$) influenced the crops biomass production. This improved the percentage of filled grains (PFG), and the thousand-grain weight (1000-GW) of treated plants by 4–5%. These improvements in growth and yield components eventually increased the grain yield production by 14%. It is concluded that partial replacement of CF with PGPR could be a viable approach to reduce the use of CF in existing rice cultivation systems. Furthermore, the approach has potential as a sustainable technique for rice cultivation.

**Keywords:** plant-growth-promoting bacteria; dry matter partitioning; harvest index; rice cultivation methods; rice; sustainable agriculture

## 1. Introduction

Rice (*Oryza sativa*) is the principal dietary component for more than 50% of the world's population [1]. Global cereal production is expected to increase by 13% by the year 2027. Rice production is forecasted to increase by 64 metric tons (Mt) to 562 Mt to satisfy the dietary calorie requirements of the population [2]. For decades, conventional rice production has relied heavily on the use of chemical fertilizer (CF) to obtain the desired crop growth and yield output. The expected increase in rice requirement will further increase the need for CF during the cultivation phase for rice crop. This raises concern as plants do not absorb all nutrients supplied by CF. Plants absorb essential macronutrients such as phosphate ions ($HPO_4^{2-}$ and $H_2PO_4^{-}$), nitrate ($NO_3^{-}$), and ammonium ($NH_4^{+}$). Pollution is known to occur when these elements discharge through leaching, mineralization, or volatilization, a situation that is further amplified through rapid degradation of the soil structure during the intensive rice cropping cycle [3–5].

Studies that have investigated the use of microorganisms as an alternative to conventional methods of supplying nutrients to plants have recently gained widespread interest [6–9]. Past experiments revealed complex interactions between these microorganisms and plants that may conceivably allow reduction in the use of CF in rice production [10,11].

Plants have a limited ability to genetically adapt to rapid environmental changes (heat, drought, toxins, or insufficient nutrients) due to their short life cycles. To assist them, plants have complex interactions with microorganisms present in their rhizosphere [12–16]. Most microorganisms have no direct importance for plant growth and vigor. Nevertheless, there are a few microbe communities that are beneficial for plants. These microorganisms were termed plant-growth-promoting rhizobacteria (PGPR) by Kloepper and Schroth [17]. PGPR is known to stimulate the synthesis of plant growth regulators, phytohormones, and many biologically active substances in the host plant. Moreover, previous studies showed that specific species have the potential to accrue and alter nutrients for plant uptake [18–22].

Nguyen et al. [23] demonstrated that replacement of 50% of the required nitrogen (N) fertilizer with two *Bacillus* spp. and an *Azospirillus* sp. in wheat (*T. aestivum*) resulted in vigorous vegetative growth with an increased biomass weight. Similarly, Yang et al. [15] applied *Sphingomonas* to *Dendrobium officinale* and observed increases of 8.6% in the plant culm structure and 7.5% in its fresh weight. Castanheira et al. [12] also tested the same species of bacteria by applying three different strains along with N and phosphorous (P) supplements to ryegrass (*Lolium* spp.) and observed improvement in the overall plant growth. These studies demonstrated that the bacterium infiltrated the plant roots and stem tissues, triggering rigorous plant growth and development. Furthermore, the plants' overall physiological growth and development were more favorable when two or more PGPR strains were combined with CF. This also resulted in improvements in the crop productivity as well as a reduction in fertilizer usage. For instance, Shaharoona et al. [11] studied wheat inoculation with two species of *Pseudomonads* (*P. fluorescens* and *P. fluorescens* biotype F) with different levels of nitrogen, phosphorous, and potassium (NPK). The inoculated plants showed an increase in yield of up to 22% compared with crops cultivated with NPK alone. Besides the yield, there were significant increases in root development and the overall biomass production.

Preliminary studies have demonstrated that amending CF with PGPR reduced the demand of rice plants for synthetic fertilizer without compromising vegetative growth or the grain yield output [24–27]. Tarigan et al. [26] and Tarigan et al. [27] further reported that with the presence of PGPR, the availability of $NO_3^-$ and $NH_4^+$ was maintained throughout the crop growth cycle to a greater extent than in fields farmed using CF alone. Such information is vital for effectively reducing the use of synthetic fertilizers. However, the application of PGPR during those studies was confined to growing conditions with ample irrigation. Besides maintaining soil moisture, irrigation aids with the even dispersal of nutrients. At present, limited studies have reported on the application and response of rice cultivated with a combination of synthetic fertilizer and PGPR under alternate wetting and drying (AWD) cultivation. Therefore, a study was undertaken to investigate the effects of replacing CF with PGPR on vegetative growth, biomass production, and grain yield in rice plants cultivated under AWD cultivation. Furthermore, the study aimed to determine an optimal reduction in CF dose when incorporated with PGPR for application in rice cultivated under AWD.

## 2. Materials and Methods

### 2.1. Site Description and Experimental Design

The experiment was conducted from August to December 2018 at a research field in Taiwan (22.39° N: 34.95° E: 71 m above sea level). The soil type at the site was loamy (27% sand and 24% clay). The soil properties included a 15% wilting point, 30.5% field capacity, and 42.9% saturation. The recorded bulk density was 1.40 g/cm$^3$. Soil has a matric potential of 11.09 bar and a hydraulic conductivity of 57 mm/h [28]. The trial employed a randomized complete block design (RCBD) comprising three replicates. The

treatments consisted of a combination of N-based fertilizer and PGPR that was applied at the following ratios: 0% CF + 100% PGPR, 25% CF + 75% PGPR, 50% CF + 50% PGPR, and 100% CF + 0% PGPR, which are referred to as FP1, FP2, and FP3 in this paper, and the control (CONT). The combination of 100% CF + 0% PGPR was designated as the control treatment, as it represents the conventional dosages of synthetic nutrient fertilizers applied to rice crops under similar growing conditions.

### 2.2. Trial Establishment

Field preparation was initiated two weeks prior to transplanting. Dry tillage was conducted to a depth of 0.3 m, and the bunds were constructed up to 0.15 m in height after being compacted. Each plot was 4 m long and 1.5 m wide, amounting to a total area of 6 m$^2$. Each plot was manually levelled, and the soil was saturated before transplanting. Fourteen-day-old seedlings of the rice variety Kaohsiung 147 (KH-147) were obtained from a nursery and manually transplanted (one seedling per hill) with spacings of 25 cm between columns and 25 cm between rows (density of 16 plants per m$^2$). The experimental treatments were applied to designated plots according to the three critical growth stages: basal, active tillering, and flowering. No pesticides were applied while weeding was carried out manually.

### 2.3. Irrigation Management

Irrigation was applied immediately after transplanting and saturated field conditions were maintained for 30 days prior to implementing the irrigation interval scheme. Irrigation to a water depth of 3 cm per 7 days was used in accordance with Pascual and Wang [28]. The amount of water required to obtain this depth was calculated using the formula adopted from Brouwer et al. [29].

### 2.4. Bacterial Strains

Three combinations of bacterial strains were identified and used in this experiment. This included the bacteria species *Bacillus aryabhattai*, which was recently acquired by Dabang Protein Co. Ltd. for commercialization. Each strain was obtained from air samples at a local rice field in Taitung County using a microbiological air sampler (Coriolis M. Bertin Technologies). The device was randomly placed at six locations within a hectare of a rice paddy field. It was rested on the soil surface with the air suction tube suspended approximately 20 cm off the ground (Figure 1). Sampling at each site was carried out in 10-min intervals. This procedure was conducted in this way to draw out any bacterial life present in the air as a consequence of evaporation and/or transpiration within the rice ecosystem.

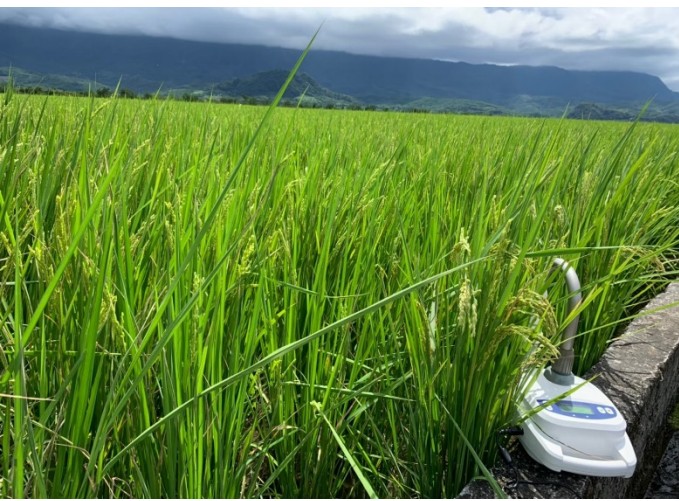

**Figure 1.** Air samples being drawn at a rice field in Taitung County, Taiwan.

The solution was brought to the laboratory, where the processes of isolation and screening were conducted. Bacterial isolation was achieved using the dilution plate technique, which involves the use of nutrient agar media. The media comprised 2.5 g $L^{-1}$ glucose, 4.8 g $L^{-1}$ lactose, and 20 g $L^{-1}$ sodium acetate. Bacterial colonies grew in the media for five days at $25 \pm 2$ °C. The procedure was repeated three times to isolate and obtain clusters of colonies each corresponding to a single strain.

The bacterial strains were selected based on characteristics such as shape, color, margins, and density. Clusters of strains with smaller populations were regarded as redundant isolates and were discarded. The process yielded pure colonies of seven bacteria strains. The purified samples were immediately sent for identification. A batch of samples was sent to the Food Industry Research and Development Institute, Hsinchu, Taiwan (ROC) for bacterial identification using matrix-assisted laser desorption/ionization time of flight (MALDI-TOF) analysis. However, three out of the seven bacterial strains from the batch could not be identified using this procedure. Thus, a second batch was sent to Mission Biotech Taipei, Taiwan (ROC) for identification using the 16S rRNA gene sequencing method. A list of the identified bacterial strains is presented in Table 1. From the seven identified bacterial strains, the three strains *B. aryabhattai*, *Burkholderia ambifaria*, and *Sphingobium yanoikuyae* were used for this study.

**Table 1.** Details of bacterial species and modes of identification.

| Entry | Mode of Identification | Name |
|---|---|---|
| PGPR_001 | 16S rRNA sequence | *Bacillus aryabhattai* |
| PGPR_002 | 16S rRNA sequence | *Burkholderia ambifaria* |
| PGPR_003 | 16S rRNA sequence | *Stenotrophomonas maltophilia* |
| PGPR_004 | MALTI-TOF | *Sphingomonas* sp. |
| PGPR_005 | MALTI-TOF | *Burkholderia caribensis* |
| PGPR_006 | 16S rRNA sequence | *Sphingobium yanoikuyae* |
| PGPR_007 | MALTI-TOF | *Paenibacillus glucanolyticus* |

Remarks: PGPR = plant-growth-promoting bacteria, 16S rRNA = 16S ribosomal ribonucleic acid, S = the Svedberg unit of measurement, and MALDI-TOF = matrix-assisted laser desorption/ionization time of flight.

*2.5. Treatments and Application Rates*

CF ($N/P_2O_5/K_2O$) was applied at a ratio of 12:18:12 at a rate of 270 kg/ha at the basal, active tillering, and panicle initiation (PI) stages [28]. A combination of all three bacterial strains was used as the PGPR treatment. The purified colonies were cultured in 1 L LB liquid medium at 25 °C for 24–36 h, and the number of bacterial colonies was then measured with a spectrophotometer at $OD_{600}$ to ensure this value was greater than 1 ($2 \times 10^9$ CFU/mL = 1.0) before application. The PGPR dosage was dispersed equally in each plot by diluting the solution in the irrigation water source and applying it to the rice plants during the scheduled seven-day irrigation period. The PGPR solution was dispensed at a rate of 200 L per hectare along with the CF doses. Both the CF and PGPR application rates were calculated based on the plot size (Table 2).

**Table 2.** Treatment description and rate of treatment application according to the plot size of 6 m².

| No. | Entry | Treatment Description | Type | CF Rate | PGPR Rate |
|---|---|---|---|---|---|
| Treatment 1 | FP1 | 0%CF + 100% PGPR | Single | 0.0 g | 120.0 mL |
| Treatment 2 | FP2 | 25%CF + 75% PGPR | Combined | 40.5 g | 90.0 mL |
| Treatment 3 | FP3 | 50%CF + 50% PGPR | Combined | 81.0 g | 60.0 mL |
| Control | CONT | 100%CF + 0% PGPR | Single | 162.0 g | 0.0 mL |

Remarks: FP = treatment; CONT = control treatment, and CF = nitrogen-based chemical fertilizer.

### 2.6. Vegetative Assessment

The vegetative assessment focused on the three growth stages of active tillering (ATS), flowering (FS), and maturity (MS). The assessment parameters included the plant height, tiller number, and leaf area index (LAI). Six plants were used for the vegetative growth assessments. Assessments were carried out for every replicate in each treatment group. The sample plants were selected from the center of the plot and were repeatedly assessed to determine the effects of treatment on crop growth (Figure 2). Plant height was measured from the base of the plant to the tip of the highest leaf, whereas the tiller numbers were quantified by manually counting the tillers. The LAI was determined using the method applied by Yoshida [30].

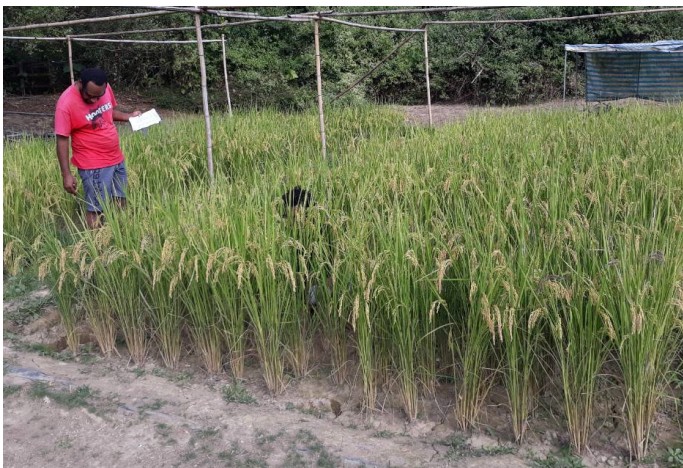

**Figure 2.** Vegetative assessment in progress at the trial site. The rice plants were approaching 100 days after transplanting.

### 2.7. Chlorophyll Content

The crop chlorophyll content was measured using a Soil Plant Analysis Development (SPAD) hand-held device (model SPAD-502, MINOLTA, Japan). The SPAD provides a rapid non-destructive approach to measure the chlorophyll content of crops in the field. The SPAD chlorophyll meter quantifies the greenness of a target plant over a specific growth period [31]. Therefore, it is an essential tool for estimating the N status of the crop, which was assessed concurrently with vegetative assessment. Like the vegetative assessment, six sample plants from each replicate of each treatment group were assessed at two-week intervals.

### 2.8. Dried biomass Weight Assessment

Dry matter assessment was carried out twice by Gendua et al. [32]. The initial assessment was carried out at the FS, and the second assessment was when the rice reached the MS. The procedure began with random selection of five sample plants from each replicate from each treatment plot. Each sample plant was carefully uprooted, and the leaves, stem, and panicles were dissected. The samples were then oven-dried at 95 °C for the first two hours. The temperature was then reduced to 65 °C where the samples were kept for 48 h. The dry weights of the different plant organs were measured separately using an electronic scale (SI-132, Excell Precision Co., Ltd., Taiwan).

### 2.9. Yield and Yield Components

The yield assessment commenced at 100 days after transplantation (DAT) when the plants reached the MS. Twelve sample plants from each replicate in each treatment group were randomly sampled and dissected to separate the panicles from the plant stalks. The samples were then labeled and oven-dried at 70 °C for 72 h. The grain yield was determined according to the method presented by Gendua et al. [32] and Akinwale et al. [33] using

the four major yield components: the number of panicles per hill (No. Panicle/hill), the number of spikelets per panicle (NS-P), the percentage of filled grain (PFG), and the 1000 grain weight (1000-GW). The assessment parameter plant dry matter weight (DMW) was obtained from the same 12 sample plants harvested at the MS for the grain yield assessment. The harvest index percentage (HI) was determined from the grain yield and DMW following the method presented by Amanullah and Inamullah [34].

### 2.10. Statistical Analysis

All data were tabulated using Microsoft Pro 2019 Excel. All statistical analyses were conducted using SPSS (Version 22, IBM, Armonk, New York, NY, USA). An analysis of variance (ANOVA) was performed, and the means were compared using Duncan's multiple range test (DMRT) at a significance level of 5%. Pearson correlation at a significance level of 5% was then used to determine the association of the grain yield with the crops' vegetative development and yield components.

## 3. Results

### 3.1. Plant Height

In terms of the plant height during ATS (44 DAT), measurements indicated no significant treatment effect (Figure 3a). However, a treatment effect was evident at the FS (72 DAT). Plants treated with FP3 (100.69 cm) produced plants with significantly greater heights ($p \leq 0.05$) than those treated with FP1 (61.17 cm). A further treatment effect was apparent when the rice plants reached the MS. The plants exposed to the control treatment achieved the highest plant heights (110.11 cm), while the respective heights of plants treated with FP3 (105.68 cm), FP2 (106.52 cm), and FP1 (104.19 cm) were all significantly lower ($p \leq 0.05$).

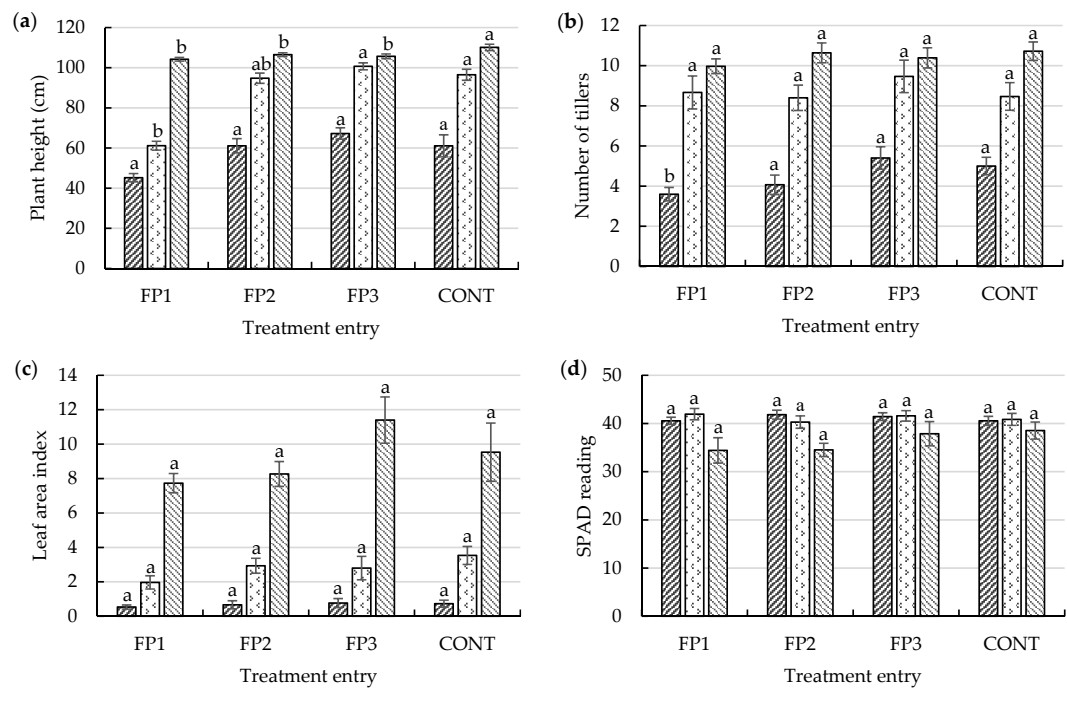

**Figure 3.** Effects of the four different treatments on the vegetative growth including chlorophyll measurements at 44 DAT (ATS), 72 DAT (FS), and 100 DAT (MS) under AWD. (**a**) Treatment effect on the plants height. (**b**) Treatment effect on the crops' number of tillers. (**c**) Treatment effect on the crops' leaf area index (LAI). (**d**) Treatment effect on the crops' chlorophyll content according to SPAD measurements: FP1 = 0% CF + 100% PGPR, FP2 = 25% CF + 75% PGPR, FP3 = 50% CF + 50% PGPR, and CONT = 100% CF + 0% PGPR. Different letters over the error bars represent statistically different values at $p \leq 0.05$ according to DMRT.

### 3.2. Number of Tillers

The number of tillers is closely related to the number of panicles and may influence the yield output if all tillers are productive [35]. Throughout the trial, the treatment effect on the number of tillers was minimal (Figure 3b). The only notable difference among the treated plants occurred during the early growth stage (44 DAT). FP3 treatment resulted in the greatest number of tillers during that period (5.4 tillers) followed by CONT (5.0 tillers), with both producing significantly more ($p \leq 0.05$) than that by FP1 treatment (3.6 tillers). As crop growth progressed to FS (72 DAT) and MS (100 DAT), no treatment effect was apparent on the tiller number of those treated plants. Overall, the average numbers of tillers produced during FS and MS were 8.75 and 8.42.

### 3.3. Leaf Area Index

The results for LAI development during the three critical growth stages of the rice plants are shown in Figure 3c. Trial assessments showed no significant effect ($p \leq 0.05$) on the LAI of plants exposed to the four treatments. FP3-treated plants had the highest LAI at the ATS (0.77) and MS (11.40), whereas FP1-treated plants had the lowest LAI throughout the experiment.

### 3.4. Chlorophyll

The chlorophyll content was measured using a SPAD meter, since this tool can be used to quantify the greenness of rice plants at any targeted growth period. Although it does not determine the quantity of N required by crops, its application in modern crop farming systems has greatly aided crop producers in the management and improvement of N fertilizer application [36]. The SPAD readings obtained from this study are presented in Figure 3d. There were no significant differences between the plants of the four treatments at the three growth stages assessed. At the ATS, the readings showed values ranging from 40 to 42 (Av. 41.27), and there was almost uniform greenness across all plants from the four treatment groups. The range of plant leaf greenness values (regardless of the treatment) remained constant until the crops reached the FS (Av. 40.91). The study recorded a reduction in plant greenness of about 18% at the MS (Av. 36.97). Although there were no statistically significant differences, the chlorophyll content in plants treated with FP2 (34.51) and FP1 (34.41) was reduced considerably at the MS compared with that of FP3-treated (37.88) and control (38.53) plants.

### 3.5. Dry Matter Partitioning

The results tabulated in Table 3 indicate the crops' peak ability to assimilate nutrients and convert absorbed nutrients to biomass following exposure to the four treatments used in this experiment. At the FS (72 DAT), the dry culm matter (SDM) was significantly affected by the treatments. Generally, all sample plants treated with a combination of CF and PGPR produced notably greater amounts of SDM than plants exposed to the control treatment. FP1-treated (10.99 g) plants generated significantly more ($p \leq 0.05$) SDM than CONT (6.07 g) plants. At the MS (100 DAT), all sample plants indicated no significant differences.

**Table 3.** The culm, leaf, panicle, and total dry matter accumulation of rice plants at the flowering and maturity stages following treatments combining chemical fertilizer and plant-growth-promoting rhizobacteria.

| Plant Dried Partition | Treatments | Growth Stage | |
| --- | --- | --- | --- |
| | | Flowering | Maturity |
| Culm dry matter (g) | FP1 | 10.99 a ± 3.26 | 16.52 a ± 4.76 |
| | FP2 | 8.71 ab ± 5.08 | 16.66 a ± 3.45 |
| | FP3 | 8.93 ab ± 4.49 | 17.83 a ± 5.12 |
| | CONT | 6.07 b ± 2.87 | 16.59 a ± 2.42 |
| | Av. | 8.68 | 16.9 |
| | $p \leq 0.05$ | * | ns |

**Table 3.** *Cont.*

| Plant Dried Partition | Treatments | Growth Stage | |
|---|---|---|---|
| | | **Flowering** | **Maturity** |
| Leaf dry matter (g) | FP1 | 2.95 a $\pm$ 0.89 | 4.55 a $\pm$ 0.67 |
| | FP2 | 2.43 ab $\pm$ 1.16 | 4.79 a $\pm$ 0.64 |
| | FP3 | 2.36 ab $\pm$ 1.12 | 4.74 a $\pm$ 1.26 |
| | CONT | 1.87 b $\pm$ 0.87 | 4.41 a $\pm$ 0.63 |
| | Av. | 2.41 | 4.62 |
| | $p \leq 0.05$ | * | ns |
| Panicle dry matter (g) | FP1 | 1.73 a $\pm$ 0.68 | 18.55 a $\pm$ 4.79 |
| | FP2 | 1.43 a $\pm$ 0.46 | 20.31 a $\pm$ 4.38 |
| | FP3 | 1.31 a $\pm$ 0.65 | 18.07 a $\pm$ 4.83 |
| | CONT | 1.36 a $\pm$ 0.73 | 17.51 a $\pm$ 2.59 |
| | Av. | 1.46 | 18.61 |
| | $p \leq 0.05$ | ns | ns |
| Total dry matter accumulation (g) | FP1 | 15.65 a $\pm$ 4.65 | 39.61 a $\pm$ 10.03 |
| | FP2 | 12.67 ab $\pm$ 6.66 | 41.75 a $\pm$ 7.14 |
| | FP3 | 12.60 ab $\pm$ 6.19 | 40.63 a $\pm$ 10.83 |
| | CONT | 9.27 b $\pm$ 4.30 | 38.52 a $\pm$ 4.60 |
| | Av. | 12.52 | 40.13 |
| | $p \leq 0.05$ | * | ns |

Remarks: Mean values followed by different letters within columns differ significantly at $p \leq 0.05$ according to DMRT. The standard deviation is presented after the $\pm$ symbol. * Significant at $p \leq 0.05$; ns: non-significant.

Similarly, sample plants treated with an amalgamation of CF and PGPR produced greater LDWs than those treated using only CF. At the FS, sample plants exposed to a single treatment of PGPR only (2.95 g) had significantly greater LDWs ($p \leq 0.05$) than those exposed to the control treatment (1.87 g). At the MS, the LDW values of all sample plants varied within 4 g, showing no significant differences among treatment groups.

No significant treatment effects were indicated for the panicle dry weight (PDW) during either the FS or the MS. As noted for the SDW and LDW, plants treated with both CF and PGPR produced slightly higher PDWs than those exposed to the control treatment. The overall total accumulated dry matter (TDW) at the FS indicated significantly different weights for the sample plants. Plants treated with FP1 (15.65 g) produced the highest TDW values, and plants with the lowest ($p \leq 0.05$) TDW values were those exposed to the control treatment (9.27 g). Sample plants treated with FP3 (12.60 g) and FP2 (12.67 g) shared almost similar TDW values but did not have TDW values significantly different ($p \leq 0.05$) from those of FP1 or CONT plants. The results showed no statistical differences ($p \leq 0.05$) in TDW values in the sample plants at the MS. However, plants treated with PGPR had slightly greater biomasses than those exposed to the control treatment (38.52 g).

*3.6. Dry Matter Partition Percentage*

The percentages of assimilates partitioned into the culm, leaf, and panicle during the FS and MS are detailed in Table 4. The portions of dry matter partitioned to the culm of plants treated with FP3 (70.36%) and FP1 (70.28%) were the highest ($p \leq 0.001$) at the FS. The percentage of dry matter partitioned to the culm of plants treated with FP2 (68.07%) was statistically different from that of FP3 and FP1. Plants that underwent the control treatment (65.11%) had the least partitioned dry matter. There was no difference in the portion of dry matter partitioned into the leaves of plants in the four treatment groups at the FS. However, there was a significant difference ($p \leq 0.05$) in the percentage of dry matter partitioned to the panicles at the FS. Plants exposed to the control treatment (14.31%) obtained the highest portions of dry matter in the panicles. FP3 (10.73%) and FP1 (10.89%)

were the treatments associated with the least significantly different ($p \leq 0.05$) portions of dry matter partitioned to the panicles.

**Table 4.** Percentages of dry matter partitioned to the panicles, leaves, and culms of sample rice plants at the flowering and maturity stages following treatment with different combination of chemical fertilizer and plant-growth-promoting rhizobacteria.

| Growth stage | Treatments | Percentage of Partition (%) | | |
| --- | --- | --- | --- | --- |
| | | Culm | Leaf | Panicle |
| Flowering | FP1 | 70.28 a | 18.81 a | 10.89 b |
| | FP2 | 68.07 b | 19.65 a | 12.27 ab |
| | FP3 | 70.36 a | 18.91 a | 10.73 b |
| | CONT | 65.11 c | 20.57 a | 14.31 a |
| | Av. | 68.46 | 19.48 | 12.05 |
| | $p \leq 0.05$ | *** | ns | * |
| Maturity | FP1 | 41.42 ab | 11.68 a | 46.91 ab |
| | FP2 | 39.87 b | 11.63 a | 48.50 a |
| | FP3 | 43.72 a | 11.66 a | 44.62 b |
| | CONT | 43.07 a | 11.47 a | 45.46 ab |
| | Av. | 42.02 | 11.61 | 46.37 |
| | $p \leq 0.05$ | * | ns | * |

Remarks: Mean values followed by different letters within columns differ significantly at $p \leq 0.05$ according to DMRT. * Significant at $p \leq 0.05$; ns: non-significant; *** Significant at $p \leq 0.001$; ns: non-significant.

### 3.7. Yield Components

The assessment showed no significant treatment effect ($p \leq 0.05$) on the No. Panicle/hill or NS-P for the sample plants (Table 5). At the MS, all sample plants had an average of 10 panicles per hill and an average of 125.75 grains per panicle. However, the replacement of CF with PGPR did influence both the PFG and the 1000-GW. The most significant increase ($p \leq 0.05$) in the PFG occurred for plants treated with FP2 (83.99%), followed by those treated with FP1 (83.81%). Sample plants exposed to the control treatment (79.07%) generated the least PFG in this study. The variation among the thousand rice grain weights was highly significant ($p \leq 0.01$). In general, plants exposed to PGPR produced heavier rice grains than those treated with only CONT (25.74 g) in the order of FP3 (27.18 g), FP2 (26.93 g), and FP1 (26.50 g).

**Table 5.** Number of panicles, number of spikelets per panicle, percentage of filled grains, and 1000 grain weights of rice plants treated with different combinations of chemical fertilizer and plant-growth-promoting rhizobacteria.

| Entry | Number of Panicles Per Hill | Number of Spikelets per Panicle | Percentage of Filled Grains (%) | 1000 Grain Weight (g) |
| --- | --- | --- | --- | --- |
| FP1 | 9.53 a ± 2.08 | 123.81 a ± 22.61 | 82.81 a ± 5.73 | 26.50 a ± 1.51 |
| FP2 | 10.17 a ± 2.77 | 128.43 a ± 16.67 | 83.99 a ± 6.88 | 26.93 a ± 1.37 |
| FP3 | 10.08 a ± 2.95 | 126.61 a ± 18.01 | 81.99 ab ± 6.33 | 27.18 a ± 1.82 |
| CONT | 10.11 a ± 2.71 | 124.16 a ± 25.46 | 79.07 b ± 6.77 | 25.74 b ± 1.66 |
| Av. | 9.97 | 125.75 | 81.96 | 26.59 |
| $p \leq 0.05$ | ns | ns | * | ** |

Remarks: Mean values followed by different letters within columns differ significantly at $p \leq 0.05$ according to DMRT. The standard deviation is presented after the ± symbol. * Significant at $p \leq 0.05$; ** Significant at $p \leq 0.01$; ns: non-significant.

### 3.8. Grain Yield

The rice grain yield was determined using four major components (No. Panicles/hill, NS-P, PFG, and the 1000-GW) as shown in Tables 5 and 6. Plants treated with FP2

($469.46$ g m$^{-2}$) delivered the highest grain yield and this was statistically ($p \leq 0.05$) higher than that of plants treated according to the conventional method ($401.80$ g m$^{-2}$). As observed during the assessment of PFG and 1000-GW, all rice plants exposed to PGPR treatment showed improvements in their respective grain yield outputs.

**Table 6.** Grain yield, dry biomass weight, and harvest index percentage of rice plants treated with different combinations of chemical fertilizer and plant-growth-promoting rhizobacteria.

| Entry | Grain Yield (g m$^{-2}$) | Dry Biomass (g m$^{-2}$) | Harvest Index (%) |
|---|---|---|---|
| FP1 | 412.21 ab $\pm$ 114.54 | 843.94 a $\pm$ 194.29 | 48.69 ab $\pm$ 5.42 |
| FP2 | 469.46 a $\pm$ 138.66 | 911.07 a $\pm$ 233.62 | 51.34 a $\pm$ 5.08 |
| FP3 | 445.86 ab $\pm$ 115.06 | 929.73 a $\pm$ 221.66 | 47.90 bc $\pm$ 4.53 |
| CONT | 401.80 b $\pm$ 116.47 | 891.69 a $\pm$ 207.23 | 45.34 c $\pm$ 9.82 |
| Av. | 432.33 | 893.94 | 48.09 |
| $p \leq 0.05$ | * | ns | ** |

Remarks: Mean values followed by different letters within columns differ significantly at $p \leq 0.05$ according to DMRT. The standard deviation is presented after the $\pm$ symbol. * Significant at $p \leq 0.05$; ** Significant at $p \leq 0.01$; ns: non-significant.

*3.9. Dry Biomass Weight*

The dry biomass weight (DMW) values presented in Table 6 were collected from the same sample plants from which the grain yield was assessed (Table 5). The assessment showed that the DMW values of the treatment plants were not significantly different ($p \leq 0.05$). The treatment group with the highest DMW was FP3 ($929.73$ g m$^{-2}$), followed by FP2 ($911.07$ g m$^{-2}$). The lowest DMW was found in plants treated with FP1 ($843.94$ g m$^{-2}$).

*3.10. Harvest Index*

The crop harvest index percentage (HI) is determined using the grain yield and DMW (Table 6). Our assessment indicated that the HI values were highly significantly different ($p \leq 0.01$) among treatment groups. Plants exposed to FP2 (51.34%) had statistically higher HI values than those treated with FP3 (47.90%) and the control treatment (45.34%). Overall, plants exposed to the control treatment obtained the lowest HI values in this study.

*3.11. Correlation between the Grain Yield and Crop Vegetation*

The relationships among the vegetative parameters and grain yield production were determined using Pearson correlations (Table 7). The analysis showed that the tiller number and LAI were the only two vegetative parameters with correlations that varied significantly with the grain yield. Grain yield production was positively correlated ($p \leq 0.001$) with the LAI (r = 0.65 ***), whereas there was a negative correlation ($p \leq 0.05$) between the grain yield and tiller number (r = $-0.22$ *).

**Table 7.** Correlations of the plant height, number of tillers, leaf area index, and chlorophyll content with the grain yield.

| | Vegetative Parameters | | | |
|---|---|---|---|---|
| | Plant ht. | Tiller No. | LAI | Chlorophyll |
| Grain yield | 0.14 ns | $-0.22$ * | 0.65 *** | $-0.11$ ns |

* Significant at $p \leq 0.05$; *** Significant at $p \leq 0.001$; ns: non-significant.

*3.12. Correlations between the Grain Yield and Yield Components*

Table 8 shows the correlations between the main yield components and the grain yield output. Our analysis indicated that the grain yield and the No. Panicle/hill attained a significant negative correlation (r = $-0.21$ *) corresponding to the relationship between tiller numbers and the grain yield (Table 7). Both the NS-P (r = $-0.12$ ns) and PFG (r = $-0.08$ ns) were found to have negative correlations with the grain yield. However, neither had a

significant influence on the grain yield output. The relationship between rice grain yield production and the DMW was shown to be significantly positive. This indicates that an increase in biomass production may be linked with an increase in the crop's ability to increase grain yield production.

**Table 8.** Correlations of the yield components, dry matter weight, and harvest index of all assessed parameters with the grain yield.

| | Yield Component Assessment | | | | | |
|---|---|---|---|---|---|---|
| | No. Pani-cle/Hill | NS-P | PFG | 1000-GW | DMW | HI (%) |
| Grain yield | −0.21 * | −0.08 ns | −0.14 ns | 0.23 * | 0.22* | −0.14 ns |

* Significant at $p \leq 0.05$; ns: non-significant.

## 4. Discussion

According to the literature, the three bacteria species *B. aryabhattai, B. ambifaria*, and *S. yanoikuyae* are considered PGPR based on their interactions with and beneficial influences on host plants. The bacterial strain *B. aryabhattai* belongs to the *Bacillus* genus of the Bacillaceae family. This genus boasts some well-known PGPR species [19,37–39]. Park et al. [40] found that the strain was able to trigger production of the phytohormone abscisic acid (ABA). The bacterial strain *B. ambifaria* belongs to the *Proteobacteria* genus. Similar to *Bacillus*, it contains some known PGPR strains used in the field of agriculture [10,19,41–44]. Parra-Cota et al. [10] showed that *B. ambifaria* has the potential to be used as a PGPR. The study found that the bacteria strain was able to improve the vegetative growth and yield production of two species of amaranth (*Amaranthus cruentus* and *A. hypochondriacus*). The bacterial strain *S. yanoikuyae* belongs to the *Sphingomonas* genus and has medical, industrial, and agronomical importance [45–47]. According to Yang et al., the strain was found to promote rigorous cell wall development along the culm of *Dendrobium officinale*. A study by Hoo et al. [48] determined that the bacterial strain was able to restore magnesium ion ($Mg^{2+}$) levels along the rhizoplane for plant cell growth.

Throughout the growth cycle, the effects across treatments on vegetative parameters were minimal. A notable effect of treatment was observed in plant height, particularly at the FS and MS (Figure 3a). The height of KH-147 at maturity ranges 90–125 cm. According to the International Rice Research Institute (IRRI) [49], it is intermediate in height. This enables maximum exposure of the crop's vegetative canopy to direct sunlight for photosynthetic activities. Rice varieties with tall plant heights are disadvantaged in some situations as they are vulnerable to lodging. An increased plant height can also result from excess N fertilizer application [50]. N is fundamental for plant growth and development; however, it also encourages rapid elongation of the cell walls at the culm structure. It may weaken the resilience of crops to other forms of biotic and abiotic stress [51]. As such, lodging can lead to severe crop loss and poor grain quality [52,53]. Regardless, plant height is still an important trait that affects the potential rice yield [54].

The average plant height at the MS was 105.46 cm (Figure 3a). Application of a high concentration of CF during the early growth stages may have led to increased heights in control plants (110.11 cm). Plants treated with amended doses had slower growth paces. The results indicate that plants exposed to PGPR could have accrued sturdy cells, resulting in slow growth. Previously, studies showed that plants inoculated with *Burkholderia* spp. and *Sphingomonas* spp. have thick culms and robust cell structures [10,15]. The presence of bacterial strains in plant tissues also improved the plants metabolic processes, which eventually improves their physiological growth [12].

The replacement of CF with PGPR showed no effects on tiller growth during the HS and MS (Figure 3b). The only notable effect of treatment on tiller production occurred during the ATS period. Both CONT (5.33) and FP3 (5.39) produced an average of five tillers at ATS, while the effect of a high CF input at the early active growth stage was

visible. This may be advantageous but could also result in the plants being vulnerable to pest infestation and lodging [51]. Sample plants exposed to CF responded swiftly to the nutrients supplied during the basal and ATS phases, but the number of tillers then subsided once treatment with CF ceased. Plants treated with CONT produced the highest number of tillers at the FS (9.56) and MS (10.72). However, no statistical differences in the tiller number were observed between treatments during the respective growth stages. According to Nguyen et al. [23] and Akram et al. [55], the tiller number of monocot plants such as rice is less (significantly) affected by most treatments than the crop height and other vegetative parameters. Furthermore, it is also possible that plants treated with some amount of PGPR may form resilient cell structures and have better metabolic processes along the culm area [10,15]. As such, the plants have a better chance of withstanding external stresses, thus ensuring their survival and enhancing plant productivity.

Assessment of the LAI at different growth stages is paramount, as it can be used to predict the potential yield [56,57]. An increase in the LAI of rice plants can result in an increase in their ability to intercept light [34]. The results showed no significant difference ($p \leq 0.05$) in the LAI of treated plants in this study. Throughout the initial growth stage, rice plants treated with a combination of CF and PGPR attained higher SPAD readings than those of CONT rice plants (40.56). The average leaf greenness of plants treated with FP2 (41.83) was highest at the ATS and then matched by FP1 (41.94) at the FS. From ripening to maturity, CONT (38.53) had the highest SPAD reading. The chlorophyll content fluctuated during the ATS and FS. Plants with high levels of chlorophyll during these two periods had greater photosynthetic potential, a critical factor in rice grain development. This theory was proposed based on the biomass production and dry matter partitioning observed in this study (see Tables 3 and 4). According to Tian et al. [58], a sufficient LAI is necessary for dry matter production. However, with a reduction in the amount of CF, the insignificant difference between the LAI and crop greenness demonstrates the potential of PGPR to adequately maintain typical crop growth throughout the crop's life cycle. Rice grain yield is a result of the combination of various processes, such as photosynthetic activity from the crop canopy and the portioning and conversion of assimilates to biomass [59–61]. Besides growth maintenance, the presence of PGPR may have contributed to the improvement of yield components (see below).

As observed in Table 3, the crops responded to the presence of PGPR with a significant increase ($p \leq 0.05$) during the early growth stages, as reflected by the SDW and LDW. The increased dry weights of these respective plant organs indicate the effects of *B. ambifaria* and *S. yanoikuyae* in the PGPR treatment. Plants inoculated with *Sphingomonas* spp. and *Burkholderia* spp. showed increased growth. The improvement in the vascular system by these PGPR species aids in the transport of water and nutrients from the plant roots to the aboveground plant biomass [10,12,15,62]. Another contributing strain is *B. aryabhattai*. Previous studies reported that it could positively influence both plant DMW and grain yield production [23,63]. According to Radhakrishnan et al. [64], the *Bacillus* spp. is a unique and economically useful PGPR strain. It is known to vary in strength and effectiveness and may trigger multiple effects when in contact with the plant. It could break down soil carbon and mineralize insoluble soil nutrients for plant uptake. The availability of N and P ions for uptake from soil is then increased along with traces of minor elements, such as magnesium ($Mg^{2+}$), iron ($Fe^{2+}$), zinc ($Zn^{2+}$), and chelates, when inoculated with *Bacillus* spp. The activities of these microorganisms may have improved the crop biomass production, as indicated by the increase in the crops dry weight.

The PFG for FP2 plants (83.99%) was significantly greater than that of control plants (79.07%). According to Castanheira et al. [12] and Elekhtyar [65], the microbes present in the plant tissue may assist the crop in the mobilization of nutrients and carbohydrates accumulated during the grain-filling period: the higher the PFG, the greater the rice grain yield at the maturity stage. The 1000-GW displayed a similar outcome. The results show that plants treated with a combination of fertilizer and PGPR had a significantly increased ($p \leq 0.05$) 1000-GW. The results correspond to those of Alam et al. [66], who found

similar outcomes when testing different combinations of bacterial strains (*Azotobacter* spp., *Bacillus* spp., *Enterobacter* spp., and *Xanthobacter* spp.). Plants exposed to FP3 (27.18 g), FP2 (26.93 g), and FP1 (26.50 g) gained statistically higher 1000-GW values than plants under the CONT (25.74 g) treatment. The results indicate that rice plants inoculated with PGPR have ample plant heights, low numbers of tillers, and consistent chlorophyll contents (SPAD reading) during the critical growth stages, contributing to statistically higher PFG and 1000-GW values.

The grain yield is the final indicator of crop performance for the different treatments. The combined effects of PGPR and CF in FP2 produced a 14.41% increase in performance compared with plants treated with the conventional method (CONT). The results from this study are similar to those obtained by Parra-Cota et al. [10], Shaharoona et al. [11], and Islam et al. [67]. The results for the yield component, PFG, and 1000-GW showed evidence that the presence of PGPR improved the crop performance by 4–5% compared with the control treatment (Table 5). According to Xie et al. [68] and Huang et al. [69], there is a close relationship between the two yield components. Both the grain size and grain filling rate were used to determine the 1000-GW.

The rice yield is known to be regulated by both genetic and external environmental factors [54,70,71]. The number of tillers produced by a rice plant is closely related to the number of panicles per hill [35]. However, the negative correlation indicates that high tiller production may significantly reduce the grain yield, which is a trait limited to the specific rice variety. Presumably, the conventional CF dosage may cause plants to prioritize tiller production well beyond the FS while failing to shift their efforts towards grain development. As per the results shown in Figure 3b, plants treated with all CF doses showed an increase in their respective tiller numbers, except for those treated with FP1. However, plants treated with PGPR had significant increases ($p \leq 0.05$) in their PFG and 1000-GW (Table 3). As explained by Castanheira et al. [12], the microbial strains present inside a plant's tissues aid in efficiently mobilizing nutrients and carbohydrates, which may have contributed to the improvements in the two yield components.

The LAI indicates a crop's ability to intercept radiation and precipitation, convert energy, and maintain the water balance [72,73]. As such, it is an agronomic parameter commonly assessed for predicting grain production [57]. The correlation results indicate accession in the LAI significantly improves the grain yield (r = 0.65 ***). An increase in the LAI will improve grain yield production [74–76]. The canopy of leaves over a plant allows much-needed sunlight to be intercepted to drive the plant growth and rice grain development. However, the extent to which these processes occur also depends on the height of the plant. The heights of rice plants in this study demonstrated intermediate characteristics that may have contributed towards the positive correlation between the grain yield and LAI (Figure 3a).

The correlation between the grain yield and the number of panicles per hill indicates a reduction in the grain yield output. This may be based on the genetic traits of the rice variety rather than on the treatment effect. The No. Panicle/hill is a vital grain yield component, as it reflects the effects of both the environment and cultivation practice used [77]. A literature review suggested that improvements in crop management are generally associated with an increase in the No. Panicle/hill [78,79].

The correlation between grain yield and the 1000-GW (r = 0.23 *) indicated a positive relationship. However, according to Yang and Zhang [80], the yield component 1000-GW remains constant across different environments for some rice cultivars. In our study, the application of PGPR influenced the 1000-GW (Table 5). This is an essential contribution toward an improvement in grain yield. These improvements can be attributed to the combination of CF and PGPR, thus further confirming an advantage of incorporating them both into the rice farming systems.

This study observed a significant treatment effect ($p \leq 0.05$) on biomass production across all treatment groups during the FS (Table 3) and particularly for plants exposed to PGPR treatment. According to Wu et al. [59] and Katsura et al. [81], the grain yield



increases as the biomass weight increases. Katsura et al. [81] further mentioned that dry matter accumulation before the FS greatly influences the grain yield output. Although our analysis indicates no statistical difference at the MS, an increase in the total DMW at FS contributed considerably to the respective PFG ($p \leq 0.05$) and 1000-GW ($p \leq 0.01$) values (refer to Tables 3 and 5), which ultimately resulted in higher grain yields (Table 6). The positive interaction between the grain yield and DMW (r = 0.22 *) further justifies this hypothesis (Table 8).

The HI values in this study correspond to those reported by Parra-Cota et al. [10]. A combination of both N fertilizer and PGPR was able to increase the crop HI. The assessment of plant HI indicated a highly significant difference ($p \leq 0.01$) between treatments (Table 6). However, its interaction with grain yield production showed no significant effect ($p \leq 0.05$) (Table 8). Regardless, rice grain yield production is the outcome of the biomass yield and HI, and if either or both are affected by the replacement of CF with PGPR, an increase in grain yield will occur [30,61,82]. This study only observed an increase in biomass weight in plants treated with PGPR. These improvements in biomass production may have influenced the HI. The results of this study show that PGPR aid rice plant growth as it is being reflected the by increased in biomass production thereby improving grain production.

## 5. Conclusions

This study concludes that combining CF and PGPR under the AWD cultivation system can significantly and positively influence crop growth, biomass production, and grain yield. The results suggest that the replacement of 50% or more of the CF content with PGPR is sufficient to maintain normal crop growth and development. The critical growth period, in which treatment application is essential, is from the early growth stage to the crop flowering stage. The assessment of dry matter weight and partitioning suggests that the presence of PGPR helps to promote vegetative growth during these critical periods. Both the percentage of filled grain and the thousand-grain weight were shown to significantly increase grain production. Hence, it is possible to amalgamate the use of PGPR into the conventional method of cultivating rice without compromising the yield output. Much is yet to be understood about these beneficial bacteria species and their influences on rice production. However, the current study shows the potential of PGPR and serves as a practical reference for reducing the use of synthetic fertilizers in rice cultivation. Not only is its incorporation into existing rice cultivation methods an environmentally friendly strategy but it has also been demonstrated to be an advantageous approach for rice cultivation in terms of productivity.

**Author Contributions:** Conceptualization, C.K.K.; methodology, C.K.K.; validation, Y.-T.J. and Y.-M.W.; formal analysis, C.K.K.; investigation, C.K.K.; resources, Y.-T.J. and Y.-M.W.; data curation, C.K.K.; supervision, Y.-T.J. and Y.-M.W. The published version of this manuscript was revise and agreed upon by all authors. All authors have read and agreed to the published version of the manuscript.

**Funding:** This research was co-funded by the International Master Program in Soil and Water Engineering (IMPSWE) under the supervision of Yu-Min Wang and COA grant no. 109AS-7.1.3-IE-b2.

**Institutional Review Board Statement:** Not applicable.

**Informed Consent Statement:** Not applicable.

**Data Availability Statement:** Data presented in this article is available at the NPUST library database.

**Acknowledgments:** The author would like to thank ICDF-Taiwan for providing the study opportunity and DaBomb Protein Co. Ltd., Taiwan for expressing interest in this research and aiding in the acquisition of a bacterial strain. We also give thanks to the staff and my fellow colleagues from the Department of Tropical Agriculture and International Cooperation (DTAIC), International Master Program in Soil and Water Engineering (IMPSWE), Department of Biological Science and Technology and National Pingtung University of Science and Technology for facilitating all research activities.

**Conflicts of Interest:** There are no conflict of interest among the authors.

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
