# Peer review of "Advantages of Amending Chemical Fertilizer with Plant-Growth-Promoting Rhizobacteria under Alternate Wetting Drying Rice Cultivation"

_agriculture, doi:10.3390/agriculture11070605_

Round 1
Reviewer 1 Report
The manuscript submitted to me for revision examines the effects of co-fertilization of chemical fertilizer and PGPR inoculants on rice growth and production. It is an important subject; moreover, rice is a traditional crop with great importance in Asia. The work possesses the structure recommended by the journal.
In the beginning, the abstract is well structured and gives the necessary information to the readers but is longer than the journal requirements and needs to be rewritten. The keywords are well selected, yet I would also add “rice“.
The introduction covers all experimental parts of the manuscript, and the literature references are well selected and sufficiently detailed.
The Materials and methods section is developed in detail, understandable, and could be easily reproduced. The authors cultivated rice employing RCBD with four treatments and three replicates each one. It is slightly confusing for me that “PGPRs used for the study were isolated from “air sample”. The authors have to explain that and how they knew that the strains possess plant growth-promoting abilities. It is not described. Furthermore, different plant parameters were assessed: chlorophyll content, dry biomass, yields, etc. Finally, the statistical analysis was realized using SPSS and Duncan’s multiple range tests and Pearson correlation.
The results are statistically analyzed and clearly presented. They include data about diverse aspects of plant growth, development, and yield. I think that to complete the study, data about the inoculants' development in soil and their recovery is missing. These would highly increase the manuscript’s quality.
However, the extensive discussion section is well developed
It is not shown what “DAT” does mean in the first time it appears, figure 1.
The authors have to unify the way of writing the symbol for temperature (oC) in the whole document.
Line 174: The sentence should be revised because of the use of words with the same root.
Line 423: The sentence needs to be reconsidered.
Line 492: The sentence needs to be rewritten.
The obtained results and discussion support the conclusions.
References need to be rewritten according to the Instructions to authors of the Agriculture.
Author Response
Greetings,
Please find below my response to your comments/suggestions. Also attached is the edited manuscript.
Kind regards,
Kobua
The manuscript submitted to me for revision examines the effects of co-fertilization of chemical fertilizer and PGPR inoculants on rice growth and production. It is an important subject; moreover, rice is a traditional crop with great importance in Asia. The work possesses the structure recommended by the journal.
In the beginning, the abstract is well structured and gives the necessary information to the readers but is longer than the journal requirements and needs to be rewritten. The keywords are well selected, yet I would also add “rice“.
- Minor editing was done to the Abstract section. The research objectives were rephrased (Lines 17-23).
- Minor changes to the keyword. The word “alternate wetting and drying” was replaced with “rice cultivation” to avoid the repetition of the title description. The word “rice” was also added as suggested (Line 36-37).
The introduction covers all experimental parts of the manuscript, and the literature references are well selected and sufficiently detailed.
- A hypothesis was derived based on past research.
- The objectives were rephrased.
- Please refer to Lines 83-92.
The Materials and methods section is developed in detail, understandable, and could be easily reproduced. The authors cultivated rice employing RCBD with four treatments and three replicates each one. It is slightly confusing for me that “PGPRs used for the study were isolated from “air sample”. The authors have to explain that and how they knew that the strains possess plant growth-promoting abilities. It is not described. Furthermore, different plant parameters were assessed: chlorophyll content, dry biomass, yields, etc. Finally, the statistical analysis was realized using SPSS and Duncan’s multiple range tests and Pearson correlation.
- The bacterial strain sampling and identification process were revised. Please refer to Lines 127-152.
- Based on literature the bacterial strains used in this study were previously used in some agricultural studies. Please refer to Line 394-407.
The results are statistically analyzed and clearly presented. They include data about diverse aspects of plant growth, development, and yield. I think that to complete the study, data about the inoculants' development in soil and their recovery is missing. These would highly increase the manuscript’s quality.
However, the extensive discussion section is well developed
- The Discussion section was expanded to discuss the types of bacteria species used in this study as requested by reviewer 2 & 3. Refer to Lines 394-407.
It is not shown what “DAT” does mean in the first time it appears, figure 1.
- The abbreviation has been edited (Line 175) and made mention in Line 192.
The authors have to unify the way of writing the symbol for temperature (oC) in the whole document.
- The correct symbol for temperature was updated.
Line 174: The sentence should be revised because of the use of words with the same root.
- The word “plant” was omitted out. Refer to Line 211.
Line 423: The sentence needs to be reconsidered.
- The higher the PFG, the better the crops can increase rice grain yields at the maturity stage. Refer to Line 477-478.
Line 492: The sentence needs to be rewritten.
- The sentence was revised. Refer to Line 549-550.
The obtained results and discussion support the conclusions.
References need to be rewritten according to the Instructions to authors of the Agriculture.
- The reference section has been revised and updated to the Taylor and Francis - American Chemical Society format.

Reviewer 2 Report
Review Comments
In the manuscript entitled “Advantages of Amending Chemical Fertilizer with Plant Growth Promoting Rhizobacteria under Alternate Wetting & Drying Rice Cultivation”, a field trial was undertaken in southern Taiwan under lowland cultivation condition aiming to (1) test the effects of amending CF with PGPR doses on rice growth and grain yields and; (2) determine the amount of recommended CF dose can be reduced when amended with PGPR so that a rational amount can be established. The treatment effect was observed among the dry matter partitioning, particularly for plants treated with a combination of CF and PGPR. Yield components; Percentage of filled grains (PFG) and thousand-grain weight (1000-GW) of FP3 and FP2 were significantly improved by 4–5% respectively. These increased grain yield production by 10–14% compared to the conventional approach. It was evident that PGPR could reduce CF inputs by 50-75% while significantly (P<0.05) improve grain yields. It is concluded that implementing amendments of CF with PGPR could be a viable approach to mitigate the use of CF in existing rice cultivation systems.
In general, the study is well-conducted and provided some important findings that could boost rice growth and yield. However, some revisions are required as shown below;
- The manuscript has many mistakes in English grammar and sentences structure, so I recommend the authors to used English proofreading service.
- The abstract should highlight the most important findings of the parameters measured in this study.
- The introduction is not discussing the review of literature and hypothesis behind the topic of this study. Please cite and discuss the recent studies and literature. Additionally, the objective of the study should be mentioned in more details at the end of the introduction section.
- Material and Methods:
- Why did you choose those concentrations in the treatments? Based on preliminary study or what?
- Number of sample or biological replicates should be mentioned for each parameter in the methods!
- Results are clear and well-represented.
- The discussion should be interpreted with the results as well as discussed in relation to the present literature.
Lines 370-371: Please correct this sentence “Previously 370 studies inoculate that Burkholderia spp. and Sphingomonas spp. are found to enlarge plant's culm 371 thickness with enhanced robust cell structure”
- Line 389: “The assessment of LAI for rice is paramount” rephrase and cite this sentence.
- Line 451: cite this sentence “LAI plays a substantial role in photosynthesis, energy conversion, and water balance provid-451 ing essential support for typical physiological growth and development in plants”.
-The conclusion section should be re-written to highlight the significant findings and recommendations of this study as well as to mentioned the future perspectives.
- The references section should be updated as per my above-mentioned suggestion
Author Response
Greetings,
Please find below my response to your comments/suggestions. Also attached is the edited manuscript.
Kind regards,
Kobua
In general, the study is well-conducted and provided some important findings that could boost rice growth and yield. However, some revisions are required as shown below;
- The manuscript has many mistakes in English grammar and sentences structure, so I recommend the authors to used English proofreading service.
- Will consider this option after this submission.
- The abstract should highlight the most important findings of the parameters measured in this study.
- The objectives were rephrased (Line 18-22).
- The introduction is not discussing the review of literature and hypothesis behind the topic of this study. Please cite and discuss the recent studies and literature. Additionally, the objective of the study should be mentioned in more details at the end of the introduction section.
- A hypothesis was derived based on past research.
- The objectives were rephrased.
- Please refer to Lines 82-92.
- Material and Methods:
- Why did you choose those concentrations in the treatments? Based on preliminary study or what?
- The treatments derived from past studies. Please refer to Lines 82-92 or reference number 23-26 for supporting details.
- Further elaboration on the bacterial sampling and identification procedure (between Lines 127 and 153).
- Number of sample or biological replicates should be mentioned for each parameter in the methods!
- The sample numbers are indicated as follows;
- Sampling of air sample = 6 (Line 131)
- Vegetative assessment (Plant height, tiller number, LAI [leaf No. + leaf length (cm) + leaf width (cm)] = 6 (Line 169-170)
- Chlorophyll (SPAD) assessment = 6 (Line 181-182)
- Dry biomass weight = 5 (Line 186)
- Yield and yield component (including DMW & HI) = 12 (Line 192-193)
- Results are clear and well-represented.
- The discussion should be interpreted with the results as well as discussed in relation to the present literature.
- Minor revisions being done with added citation. Please refer to Lines 394-529.
Lines 370-371: Please correct this sentence “Previously 370 studies inoculate that Burkholderia spp. and Sphingomonas spp. are found to enlarge plant's culm 371 thickness with enhanced robust cell structure”
- The sentence has been revised (Lines 423-4424).
- Line 389: “The assessment of LAI for rice is paramount” rephrase and cite this sentence.
- The sentence has been rephrased with citations added (Lines 441-442).
- Line 451: cite this sentence “LAI plays a substantial role in photosynthesis, energy conversion, and water balance provid-451 ing essential support for typical physiological growth and development in plants”.
- The sentences have been rephrased with citations added (Lines 506-508).
-The conclusion section should be re-written to highlight the significant findings and recommendations of this study as well as to mentioned the future perspectives.
- Minor revisions were done (Lines 555-559). According to the first and third reviewer, the conclusion section is appropriate.
- The references section should be updated as per my above-mentioned suggestion
- The reference section has been revised and updated to the Taylor and Francis - American Chemical Society format.

Reviewer 3 Report
Brief Summary
The manuscript agriculture-1248456 reports results on the evaluation of the effects of replacing and amending chemical fertilization with plant growth-promoting bacteria doses on rice crops. The subject of the study could be of interest to the readers of Agriculture and addresses one of the current major concern of agriculture. However, the manuscript needs substantial improvement. See specific comments below.
Specific comments
- Abstract: Authors presented a concise abstract, briefly stating the purpose of the research, the principal results, and major conclusions.
- Introduction: The introduction correctly places the study in a broad context and highlight why it is important. The authors clearly defined the purpose of the work. However, the specific hypotheses being tested should be reported. I would revise/remove the sentence present in L59-60 “Yet, their potential and applications in food crop production are generally unknown”. Many studies presented the potential and applications of PGPB/PGPR in food crop production. Thus, this statement is not appropriate.
- Materials and Methods: The authors described very well the agronomic aspects of the study. However, they presented with insufficient detail the methods related to the bacteria identification (16S rRNA gene sequencing and MALDI-TOF analysis) and production of inoculants (media, growth conditions, density evaluation and adjustments, broth pooling and inoculants preparation).
- Results: The results description has a good quality. The authors presented the results appropriately and concisely. However, the results related to bacterial identification are completely missing.
- Discussion: This section should be ameliorated. The authors presented too much in detail the results obtained and limited the discussion to a comparison of previous studies. I would improve this section by providing a clear discussion of the findings in the perspective of literature (describing the works with sufficient detail to allow the reader to understand the work presented) and the working hypotheses. Authors should present future research perspectives. With these results, are the inoculants ready to be commercialized? Which are the limitations and the strength of the study?
- Conclusions: The key elements that the authors present in this section are appropriate.
Minor comments
The English language should be revised in the entire manuscript for minor issues and to refine some sentences.
Abbreviations should be defined the first time they appear in the abstract; the main text and the first figure or table. Please, revise all abbreviations accordingly.
I would use the term PGPB and not PGPR, authors isolated these strains from air and not from the rhizosphere. Some of them are known to be associated with the rhizosphere, however, they did not assess the ability to inhabit the rhizosphere.
Author Response
Greetings,
Please find below my response to your comments/suggestions. Also attached is the edited manuscript.
Kind regards,
Kobua
Specific comments
- Abstract: Authors presented a concise abstract, briefly stating the purpose of the research, the principal results, and major conclusions.
- The objectives were rephrased (Line 18-21).
- Introduction:The introduction correctly places the study in a broad context and highlight why it is important. The authors clearly defined the purpose of the work. However, the specific hypotheses being tested should be reported. I would revise/remove the sentence present in L59-60 “Yet, their potential and applications in food crop production are generally unknown”. Many studies presented the potential and applications of PGPB/PGPR in food crop production. Thus, this statement is not appropriate.
- The sentence “Yet, their potential and applications in food crop production are generally unknown” was removed (Lines 63-64).
- A hypothesis was derived based on past research (Line 83-92).
- The objectives were rephrased.
- Materials and Methods:The authors described very well the agronomic aspects of the study. However, they presented with insufficient detail the methods related to the bacteria identification (16S rRNA gene sequencing and MALDI-TOF analysis) and production of inoculants (media, growth conditions, density evaluation and adjustments, broth pooling and inoculants preparation).
- The bacterial identification process has been revised with details related to bacterial identification (Lines 127-153, 156-159)
- Results:The results description has a good quality. The authors presented the results appropriately and concisely. However, the results related to bacterial identification are completely missing.
- The bacterial identification was mentioned in the Materials and Methods section between Lines 127-153. A table (Table 1) was inserted to present the lab results.
- Discussion: This section should be ameliorated. The authors presented too much in detail the results obtained and limited the discussion to a comparison of previous studies. I would improve this section by providing a clear discussion of the findings in the perspective of literature (describing the works with sufficient detail to allow the reader to understand the work presented) and the working hypotheses. Authors should present future research perspectives. With these results, are the inoculants ready to be commercialized? Which are the limitations and the strength of the study?
- Minor revisions being done with added citation. Please refer to Lines 394-543.
- One out of the 3 bacterial strain tested was acquired for commercialization. Please refer to Lines 127-129.
- Limitation/opportunities in brief, please refer to Lines 554-448.
- Conclusions:The key elements that the authors present in this section are appropriate.
Minor comments
The English language should be revised in the entire manuscript for minor issues and to refine some sentences.
Abbreviations should be defined the first time they appear in the abstract; the main text and the first figure or table. Please, revise all abbreviations accordingly.
- Included the meaning of a few abbreviations found in the Abstract, Introduction and Materials and Methods section.
I would use the term PGPB and not PGPR, authors isolated these strains from air and not from the rhizosphere. Some of them are known to be associated with the rhizosphere, however, they did not assess the ability to inhabit the rhizosphere.
- The bacterial strains are considered to be PGPR as the air samples in which the samples were collected was a few centimetres off the ground. They were then cultured and applied to the crop during the growth phase so as such referred to as PGPR. Please refer to Lines 133-134.

Round 2
Reviewer 2 Report
The authors have addressed most of review comments. However, the manuscript still has severe mistakes in English grammar and sentences. Therefore, extensive editing of English language and style are required before giving a decision on this manuscript.
Author Response
Greeting,
Please find attached a copy of the manuscript (agriculture-1248456). The manuscript was revised by MDPI English editors.
Regards,
Mr. Kobua

Reviewer 3 Report
The authors answered correctly all my previous comments. I have no further suggestions.
Author Response
Greetings,
Please find attached a copy of my manuscript (agriculture-1248456). It has been reviewed by MDPI English editors.
Regards,
Mr. Kobua
